# Asymmetric Interfaces in Epitaxial Off-Stoichiometric Fe_3+*x*_Si_1−*x*_/Ge/Fe_3+*x*_Si_1−*x*_ Hybrid Structures: Effect on Magnetic and Electric Transport Properties

**DOI:** 10.3390/nano12010131

**Published:** 2021-12-31

**Authors:** Anton S. Tarasov, Ivan A. Tarasov, Ivan A. Yakovlev, Mikhail V. Rautskii, Ilya A. Bondarev, Anna V. Lukyanenko, Mikhail S. Platunov, Mikhail N. Volochaev, Dmitriy D. Efimov, Aleksandr Yu. Goikhman, Boris A. Belyaev, Filipp A. Baron, Lev V. Shanidze, Michael Farle, Sergey N. Varnakov, Sergei G. Ovchinnikov, Nikita V. Volkov

**Affiliations:** 1Kirensky Institute of Physics, Federal Research Center KSC SB RAS, 660036 Krasnoyarsk, Russia; yia@iph.krasn.ru (I.A.Y.); rmv@iph.krasn.ru (M.V.R.); iabond@iph.krasn.ru (I.A.B.); lav@iph.krasn.ru (A.V.L.); platunov@iph.krasn.ru (M.S.P.); volochaev91@mail.ru (M.N.V.); belyaev@iph.krasn.ru (B.A.B.); spintron@hotmail.com (F.A.B.); shanidze.l.v@mail.ru (L.V.S.); michael.farle@uni-due.de (M.F.); vsn@iph.krasn.ru (S.N.V.); sgo@iph.krasn.ru (S.G.O.); volk@iph.krasn.ru (N.V.V.); 2Institute of Engineering Physics and Radio Electronics, Siberian Federal University, 660041 Krasnoyarsk, Russia; 3Krasnoyarsk Scientific Center, Federal Research Center KSC SB RAS, 660036 Krasnoyarsk, Russia; 4Synchrotron Radiation Facility SKIF, Boreskov Institute of Catalysis SB RAS, Nikol’skiy Prospekt 1, 630559 Kol’tsovo, Russia; 5REC Smart Materials and Biomedical Applications, Immanuel Kant Baltic Federal University, 236041 Kaliningrad, Russia; 6REC Functional Nanomaterials, Immanuel Kant Baltic Federal University, 236016 Kaliningrad, Russia; granitny.kot@gmail.com (D.D.E.); aygoikhman@gmail.com (A.Y.G.); 7Faculty of Physics and Center for Nanointegration (CENIDE), University of Duisburg-Essen, 47057 Duisburg, Germany

**Keywords:** iron silicide, germanium, molecular beam epitaxy, epitaxial stress, lattice distortion, dislocation lattices, FMR, Rutherford backscattering, spintronics

## Abstract

Three-layer iron-rich Fe_3+*x*_Si_1−*x*_/Ge/Fe_3+*x*_Si_1−*x*_ (0.2 < *x* < 0.64) heterostructures on a Si(111) surface with Ge thicknesses of 4 nm and 7 nm were grown by molecular beam epitaxy. Systematic studies of the structural and morphological properties of the synthesized samples have shown that an increase in the Ge thickness causes a prolonged atomic diffusion through the interfaces, which significantly increases the lattice misfits in the Ge/Fe_3+*x*_Si_1−*x*_ heterosystem due to the incorporation of Ge atoms into the Fe_3+*x*_Si_1−*x*_ bottom layer. The resultant lowering of the total free energy caused by the development of the surface roughness results in a transition from an epitaxial to a polycrystalline growth of the upper Fe_3+*x*_Si_1−*x*_. The average lattice distortion and residual stress of the upper Fe_3+*x*_Si_1−*x*_ were determined by electron diffraction and theoretical calculations to be equivalent to 0.2 GPa for the upper epitaxial layer with a volume misfit of −0.63% compared with a undistorted counterpart. The volume misfit follows the resultant interatomic misfit of |0.42|% with the bottom Ge layer, independently determined by atomic force microscopy. The variation in structural order and morphology significantly changes the magnetic properties of the upper Fe_3+*x*_Si_1−*x*_ layer and leads to a subtle effect on the transport properties of the Ge layer. Both hysteresis loops and FMR spectra differ for the structures with 4 nm and 7 nm Ge layers. The FMR spectra exhibit two distinct absorption lines corresponding to two layers of ferromagnetic Fe_3+*x*_Si_1−*x*_ films. At the same time, a third FMR line appears in the sample with the thicker Ge. The angular dependences of the resonance field of the FMR spectra measured in the plane of the film have a pronounced easy-axis type anisotropy, as well as an anisotropy corresponding to the cubic crystal symmetry of Fe_3+*x*_Si_1−*x*_, which implies the epitaxial orientation relationship of Fe_3+*x*_Si_1−*x*_ (111)[0−11] || Ge(111)[1−10] || Fe_3+*x*_Si_1−*x*_ (111)[0−11] || Si(111)[1−10]. Calculated from ferromagnetic resonance (FMR) data saturation magnetization exceeds 1000 kA/m. The temperature dependence of the electrical resistivity of a Ge layer with thicknesses of 4 nm and 7 nm is of semiconducting type, which is, however, determined by different transport mechanisms.

## 1. Introduction

Spintronic devices, which have already found their application in such uses as tunnel magnetoresistive elements of hard disk read heads and random magnetic access memory cells [1], are usually based on vertical magnetic tunnel junctions, while semiconductor spintronics [2,3] more often use planar geometry. Vertical three-layer hybrid ferromagnet (FM)/semiconductor (SC)/FM structures combine both approaches and can be promising for both vertical and planar semiconductor spintronics devices [4,5]. Still, great attention of researchers is paid to Heusler alloys [6,7,8,9,10] due to their high spin polarization of conduction electrons, such as Fe_3+*x*_Si_1−*x*_, Co_2_FeSi, Fe_2_MnSi, and Co_2_FeAl. They have a cubic crystal structure and can be relatively easily grown on standard semiconductor substrates such as Si [11], Ge [12], GaAs [13]. The use of Ge on the other hand looks very attractive from a technological point of view, since it requires lower growth temperatures than Si while remaining compatible with CMOS technologies [14,15].

Furthermore, Ge is fundamentally more promising than Si, since it has a higher electron and hole mobility and a higher spin–orbit interaction, which can control spin-dependent transport by an electric field [16,17]. Thus, the creation of spintronic devices and the subsequent implementation of spin functionality, multilayer hybrid structures with controlled magnetic and transport properties are essential to building MOSFET-type devices based on vertically stacked FM/Ge/FM structures. All this requires systematic technological and fundamental research.

The consecutive growth of multilayer epitaxial structures consisting of materials such as the semiconducting Ge and FM Fe_3_Si is a nontrivial technological problem due to the different growth temperatures of the layers [18,19]. An elevated growth temperature of the Ge layer results in an increased Ge diffusion into the silicide bottom layer and also the diffusion of Fe into the substrate and Si atoms in the opposite direction. The reduction of the Ge diffusion and two-dimensional stable epitaxial growth of Ge films on the Fe_3_Si surface up to a thickness of dozens of nanometres is achieved via the formation of diffusion barriers and terminating the Fe_3_Si surface with several monolayers of silicon [11,18,20]. In turn, the tuning of the electronic and magnetic properties of the Fe_3+*x*_Si_1−*x*_ compounds may be accomplished by changing chemical and structural order and composition [21,22] that inevitably changes the landscape of the epitaxial formation of the Fe_3+*x*_Si_1−*x*_/Ge/Fe_3+*x*_Si_1−*x*_ heterostructures.

Here, we study the formation of vertical three-layer Fe_3+*x*_Si_1−*x*_/Ge/Fe_3+*x*_Si_1−*x*_ heterostructures on a Si(111) substrate. The effect of crystal quality on structural, morphological, magnetic, and transport properties and of the thickness of individual Ge layers on magnetic anisotropy and the temperature behaviour of the electrical resistance is investigated.

## 2. Sample Synthesis and Experimental Details

Two samples (#6 and #7) were synthesized, which are three-layer structures Fe_3+*x*_Si_1−*x*_/Ge/Fe_3+*x*_Si_1−*x*_ with a thickness of germanium d_Ge_ = 4 nm and 7 nm. Iron silicide Fe_3+*x*_Si_1−*x*_ layers were grown by the molecular beam epitaxy. Iron and silicon were co-deposited from different sources in an atomic ratio close to 3:1 but enriched with iron. Knudsen cells with indirect thermal heating of the crucible are used as sources. During the synthesis of the silicide layers, the substrate temperature was maintained at 150 °C. The deposition time was chosen so that the thickness of the Fe_3+*x*_Si_1−*x*_ layers was about 7–10 nm. A semi-insulating n-type Si(111) silicon wafer with a resistivity ρ = 3000–7000 Ohm·cm (phosphorus concentration n ≈ 1 × 10^12^ 1/cm^3^) was used as a substrate to minimize a Si contribution to the electric transport in a three-layer structure. Before loading into a high-vacuum (UHV) chamber, substrates were cleaned in a solution of H_2_O_2_:NH_4_OH:H_2_O in a 1:1:1 ratio and a 5% solution of hydrofluoric acid HF, followed by thermal annealing at a temperature of 900 °C under UHV conditions (for details see [23]). As a result, we obtain the well-known Si(111) 7 × 7 surface reconstruction. The substrate was kept at 150 °C for 30 min during the growth of the Fe_3+*x*_Si_1−*x*_ film. The entire synthesis process was controlled in situ using reflected high-energy electron diffraction (RHEED). The diffraction data (Figure 1b) shows the first Fe_3+*x*_Si_1−*x*_ layer on Si(111) 7 × 7 was formed epitaxially and has a single-crystal structure.

After the first layer of the silicide was grown, the sample temperature was raised to 300 °C for 10 min. Then 3–5 nm Ge was deposited. According to the RHEED data (Figure 1c), the germanium layer also has an epitaxial, single-crystal structure. After the deposition of germanium, the substrate temperature was reduced to 150 °C for 10 min and the second silicide layer was deposited. The RHEED pattern is characterised by the A2 (Strukturbericht [23]) in <110> direction. The observed diffraction suggests that the second silicide layer was formed epitaxially and has a single-crystal structure identical to the first layer. In sample #7, the Ge thickness is a factor two larger than in #6. As a result, the RHEED pattern from the Ge layer (Figure 1f) did not change and even became slightly more pronounced compared with the thinner layer on sample #6 (Figure 1c). Despite this, the diffraction pattern for the 2nd Fe_3+*x*_Si_1−*x*_ layer of sample #7 contains only reflections in the form of diffuse Debye rings. This geometry of the diffraction pattern indicates the formation of a nanocrystalline or polycrystalline structure. RHEED patterns are satisfactorily described with the Fe_3+*x*_Si_1−*x*_(111)[0−11] || Ge(111)[1−10] || Fe_3+*x*_Si_1−*x*_(111)[0−11] || Si(111)[1−10] orientation relationships (OR). These results were obtained directly in the UHV chamber.

The microstructure of the samples was studied using atomic force (AFM) and transmission electron (TEM) microscopy. The surface morphology of the films was measured using AFM in a semicontact scanning mode (DPN 5000 device, NanoInk, Skokie, IL, USA) using probes with a curvature radius of ~6 nm (CSG30, NT-MDT SI, Moscow, Russia). AFM data processing and statistical analysis of images were carried out using the free software Gwyddion (version 2.51) and an image processing package Fiji [24]. Depth distribution of Fe, Si, and Ge of the epitaxial Fe_1−*x*_Si*_x_* alloy films were studied with Rutherford backscattering spectroscopy (RBS) at the accelerator HVEE AN-2500 (6REC Functional Nanomaterials, Immanuel Kant Baltic Federal University, Kaliningrad, Russia). The films’ cross-sections and plan view TEM lamella were made using a focused ion beam (FIB) FB-2100 (Hitachi, Tokyo, Japan) setup for electron microscopic investigations. Static and dynamic magnetic properties were investigated using Lakeshore’s 8600 Series vibration sample magnetometer (VSM) and Bruker’s ELEXSYS-E580 electron paramagnetic resonance (EPR) spectrometer (Krasnoyarsk Territorial Shared Resource Center, Krasnoyarsk Scientific Center, Russian Academy of Sciences). Resistivity and current–voltage characteristics were measured using a Keithley 2634b SourceMeter precision multimeter over a temperature range of 4.2 K to 300 K in a home-built helium flow cryostat [25].

## 3. Results and Discussion

### 3.1. Structural Properties

#### 3.1.1. Analysis of Epitaxial Orientation Relationships

In the TEM images (Figure 2) one can identify three separate layers between the Si(111) substrate and the protective layer. The thickness of each layer is 7 nm for the Fe_3+*x*_Si_1−*x*_ layers of both samples and 4 nm and 7 nm for the Ge layers of samples #6 and #7, respectively. It should be noted that in the case of sample #6 (Figure 2a,b), the brightness of the upper and lower Fe_3+*x*_Si_1−*x*_ layers is noticeably different, which is mainly caused by “diffraction contrast” [26] due to the Bragg scattering at crystallographic orientations and different electron density (mass of constituent elements) and different thickness of the specimen. In the dark field mode, the higher the Z value and the density of the material are, the darker the image is. From the diffraction pattern of the silicide layers (Figure 2c) we conclude that the Fe_3+*x*_Si_1−*x*_ epilayers have the same orientation relationship Fe_3+*x*_Si_1−*x*_(111)[0−11] || Ge(111)[1−10] || Fe_3+*x*_Si_1−*x*_(111)[0−11] with the silicon substrate Si(111)[1−10] and the formation of a chemically disordered bcc-type Fe_3+*x*_Si_1−*x*_ alloy. The superstructure reflections of a chemically ordered alloy, i.e., (111, −111), are absent. In turn, the polycrystalline nature of the upper layer of sample #7 is supported by electron diffraction measured along the [−110] and [1−21] zone axis of Si. The OR derived for them are Fe_3+*x*_Si_1−*x*_(110)[001]−8.5°||Si(111)[−110] and Fe_3+*x*_Si_1−*x*_(0−11)[0−11]−8.5°||Si(111)[1−21], which indicate differently orientated crystallites (Figure 3 (Sample #7)).

We also analysed the epitaxial stress in the epilayer and in the crystallites of the polycrystalline layer of the Fe-Si alloy. Changing the chemical composition within the trilayers based on ferromagnetic Fe*_x_*(Si_1−*y*_Ge*_y_*) alloys can be used to tune the epitaxial stress and thereby the electronic structure and, as a result, the functional properties of the material [27,28,29].

The lattice distortions δ_a,b,c_ and δ_α,β,γ_ can be regarded as additives to *a*, *b*, *c* and *α*, *β*, *γ* parameters and in the most cases result in the change of crystal symmetry [30]. By measuring interplanar distances with the X-ray diffraction (XRD) method or TEM in principle, one can determine the lattice distortions solving a given system of equations relating the interplanar distances and the lattice parameters [30]. Except for interplanar distance, TEM allows one to derive the distribution of the angular distances between the planes forming the diffraction pattern [31]. For a low-symmetry crystal system, the equations become cumbersome and, in some cases, suggesting minimal distortion may be simplified to linear forms [32]. In general, they require the application of numerical methods of solving nonlinear and transcendental systems of equations.

For the epitaxial layer of the iron–silicon alloy, we consider two representations of the crystal lattice, a cubic one and a hexagonal one. The Fe_3_Si silicide belongs to the *Fm-3m* group symmetry, and it has DO3 structure (Strukturbericht). The *Im-3m* group is used for chemically disordered alloy (A2). According to the interpretation of the diffraction pattern [31], the epi-layer has the following OR: Fe_3+*x*_Si_1−*x*_(111) [0−11]||Si(111) [1−10]. The strain [33] is 3.54%, and the area misfit is −8.35% for this epitaxial OR. Thus, the *c* lattice parameter is under the same compressive stress while the *a* and *b* parameters are slightly relaxed. The hexagonal symmetry of the Si(111) surface can cause an isotropic stress for the *α*, *β*, *γ* angles. However, other combinations of the angles and *a*, *b*, *c* distortions are possible and are discussed below. The second representation is that the hexagonal symmetry of the (111) plane allows one to use the Fe_3_Si lattice with the P3 space group, where *a*, *b* correspond to [0 −0.5 0.5] and [0.5 0 0.5] directions and *c* is [111] in the cubic lattice of the Fe_3_Si. In this case, the lattice parameters are *a*, *b* = 0.3997 and *c* = 0.979 nm (c/a = 2.449) for Fe_3_Si composition [34].

The fit of the experimental set of the interplanar distances without consideration of lattice distortions yields the following lattice parameters of sample #6 for the cubic representation, *a*, *b*, *c* = 0.56801 nm, for the hexagonal one *a*, *b*, = 0.348 nm and *c* = 1.138 nm, which closely corresponds to the Fe_80_Si_20_ alloy [35]. In turn, the diffraction patterns of sample #7 correspond to *a*, *b*, *c* = 0.57137 nm (zone axis Fe_3+*x*_Si_1−x_ [1]) and *a*, *b*, *c* = 0.56994 nm (zone axis Fe_3+*x*_Si_1−*x*_ [−111]). The corresponding chemical compositions are Fe_91_Si_9_ and Fe_86_Si_14_, respectively. The calibration was carried out based on the diffraction pattern of the silicon substrate (Figure 3) with *a* = 0.54307 nm by fitting with the RANSAC lattice procedure [31]. As shown in Figure 4, the lattice parameter *a* changes in the 0.559–0.582 nm range in ternary Fe–Ge–Si alloys. These indirectly determined lattice parameters indicate that the Fe_3+*x*_Si_1−*x*_ silicide in sample #7 may contain the germanium atoms uniformly distributed over the Fe_3+*x*_Si_1−*x*_ silicide. In the case of equal silicon content, with close to 25 at.% co-deposited, the lattice parameter of the Fe_3+*x*_Si_1−*x*_ increases due to the incorporation of Ge atoms (Figure 4). The hexagonal one shows the *a*, *b* lattice constants under a large compressive strain with a value even less than the one corresponding to Si (0.384 nm), which indirectly confirms that the lattice distortion of the epilayer should be taken into account.

#### 3.1.2. Estimation of Lattice Distortions

To estimate the lattice distortions, we applied a numerical approach assuming a uniformly distributed array of random initial parameters for different configurations of the epitaxial strain. The algorithm consists of several steps restricting the range of solutions. The first step iteratively narrows the solution range based on the function $F=\overset{n}{\underset{i=1}{\sum }}f_{i}\left(\delta_{a,b,c,\alpha ,\beta ,\gamma }\right)-P_{i}$ relating the known experimental values of *P_i_* (*d*-spacing and angles between crystallographic planes observed in the diffraction pattern) by limiting each iteration to *F* < 0.2 from the uniformly distributed array of random initial parameters (2 × 10^5^ solution combinations). *n* is the total number of experimental values; the function *f* relates the *d*-spacings and angles between crystallographic planes to the unit cell parameters [30]. The procedure repeats 2 × 10^3^ times then the restricted solution range is defined by the median of the lowest and highest values of lower and upper boundaries. The second step consists of the division of the solution range by the solutions found according to the range $\delta_{a,b,c,\alpha ,\beta ,\gamma }$ values. It is divided into positive and negative ranges. Then, the range is again narrowed with the help of the hybrid approach of simulated annealing (SA) algorithm [36] and post-minimisation by the Nelder–Mead simplex algorithm [37]. For each SA procedure, the initial parameter used was taken from the uniformly distributed array of distortion parameters within the restricted range, in a total of 10^6^ sets. The final step generates a uniformly distributed array of random initial parameters (10^7^ solution sets) within the set of previously restricted sets. It limits the solutions resulting in the 200 most minimal values of the residual function defined as standard deviation from the set of experimental values *P_i_*.

To verify the applicability of the algorithm, we considered the following lattice distortion configuration, δ_a,b_ = −0.001991, δ_c_ = 0.042908 nm, δ_α,β_ = −0.07512°, and δ_γ_ = 0.12°, for a cubic lattice by solving the direct problem and by applying our numerical approach to estimate the lattice distortions. This comparison allows to estimate the set of possible solutions within an experimental error. It can be seen from Table 1 that two solutions were found. The one that corresponds to the true values shows the minor residual function (the standard deviation from the set of experimental values). It is also noticeable that the standard deviation from the average value of the stress parameters is several times less for the solution close to the true values, which indicate more narrow solution zones and can serve as a criterion to estimate the validity of one solution among others.

Several possible solutions for the three epitaxial patterns (Figure 4) are given in Table 2. For the all-epitaxial trilayer sample #6, four configurations of the distortions were considered (listed in Table 2). As can be seen, the number of possible solutions increases while relaxing the strain configuration. It is worth mentioning that the configuration with the isotropic δ_α,β,γ_ strain shows the most prominent residual function and a different δ_a_ value in comparison with others consistent with δ_a_ = −0.001368 nm. One may conclude that the better convergence is observed for two distortion configurations which are (i) δ_a_ ≠ δ_b_ = δ_c_, δ_α_ = 0, δ_β_ ≠ δ_γ_; (ii) δ_a_ ≠ δ_b_ = δ_c_, δ_α_ ≠ δ_β_ ≠ δ_γ_ (highlighted with green Table 2). It is also seen that the solution with −δ_β_, δ_γ_ and −δ_β_, δ_γ_ are equivalent. Thus, the lattice parameters of the Fe_3+*x*_Si_1−*x*_ epilayer (hexagonal representation) are *a*, *b*, = 0.4019 (0.9844) nm and *c* = 0.9848 (0.402043) nm after applying the distortion determined, and in the nondistorted case we find c/a = 2.450. The parameter *a* equals 0.568375 (0.568574) nm for the cubic representation corresponding to 19.06 and 18.37 at.% of silicon. Lattice strain for a given Fe_3+*x*_Si_1−*x*_(111)[0−11] habit plane is 0.16%; the lattice is under compressive strain since the volume misfit between the estimated undistorted and lattices is −0.63%. In the case of sample #7 (zone axis [001]), it is not possible to assess the distortion of the *c* lattice constant. It was determined that the *a* and *b* parameters are slightly distorted while δ_γ_ reaches −1.17 degrees (highlighted with green colour), with the volume misfit with the undistorted counterpart equal to −0.0369%.

The case of zone axis [−111] is the most complex to consider since the habit plane of the silicide is not characterized by a low Miller index plane. The distortion configuration was fully relaxed to estimate all six distortion parameters, i.e., δ_a_ ≠ δ_b_ ≠ δ_c_, δ_α_ ≠ δ_β_ ≠ δ_γ_. As a result, a dozen solutions were found with similar values of the residual function F~1.79 × 10^−5^. Most of them indicate a relatively wide solution zone according to the deviation from the average value with high values of the angles’ distortions (a representative one is shown in Table 2 (Sample #7 cubic–zone axis [−111])). However, one solution is characterized by a low deviation (highlighted with green colour), which we consider a distortion close to the true value. This distortion configuration results in the volume misfit of −0.996%. Thus, all cases we considered indicate that the Fe_3+*x*_Si_1−*x*_ silicide is under compressive stress.

To further estimate the reliability of the results for the lattice distortion of the Fe–Si epilayers, the density functional theory (DFT) was applied. To mimic the epitaxial strain possible in Fe_3_Si(111)/Si(111), the Fe_3_Si hexagonal unit cell with constrained *a* = *b* = 0.384 nm and γ=120° angle lattice parameters were used. The *c*, *α*, and *β* lattice parameters and atomic positions were allowed to relax. The residual stress was estimated by calculating the stress tensor for the applied lattice distortions determined for the cubic representations of sample #6 [0−11] and #7 [−111] on the optimised off-stoichiometric Fe_80_Si_20_ and Fe_86_Si_14_ unit cell. The DFT calculations were carried out with the help of The Quantum Espresso (QE) package [38]. The electronic exchange-correlation energy was selected using the generalized gradient approximation (GGA) of the Perdew–Burke–Ernzerhof (PBE) scheme [39]. To optimize the unit cell geometry the first Brillouin zone in the reciprocal space was sampled on 8 × 8 × 2 and meshes were chosen according to the Monkhorst–Pack scheme. In all calculations, the cutoff energy E_cutoff_ was equal to 30 Ry. The optimisation of the geometry was performed until the maximum values of the forces acting on atoms were less than 10^−4^ Ry/bohr.

The optimized lattice parameters derived from the DFT calculation of the strained hexagonal lattice of Fe_3_Si are *c* = 0.99353 nm (c/a = 2.587), δ_α_ = 0.0431°, δ_β_ = −0.0432°. They are in relative correspondence with the distortion configuration of sample #6 (hexagonal, Table 2). Our calculation of the residual stress reveals that the crystal lattice of Fe_3+*x*_Si_1−*x*_ alloy in both samples is under compressive strain equivalent to 0.21 and 0.91 GPa.

The transition of the upper Fe–Si alloy layer of sample #7 into a polycrystalline morphology is a consequence of different factors. We assume that the dominant one is the interface of the intermediate Ge layer. The RHEED pattern (Figure 1) indicates that the surface of the Ge layer presents 3D islands with a typical size of 1 nm [40]. Such monocrystalline islands should be faceted enough to serve as separated centres for the formation of Fe_3+*x*_Si_1−*x*_ islands so that not only the (111)||(111) interfaces may appear. The second factor is the lattice mismatch of the Fe–Si and Ge layers. The sample #7 should have a more complex interface structure and composition on the lower Ge/Fe_3+*x*_Si_1−*x*_ and Fe_3+*x*_Si_1−*x*_/Si boundaries due to a two-times more extended exposition at elevated temperature. The Ge diffusion toward the substrate and silicon atoms in the opposite direction changes the composition and, consequently, the lattice parameters of the epilayer. An increased discrepancy in lattice misfits may promote the formation of differently orientated crystallites to relax the higher values of the epitaxial stress.

#### 3.1.3. Characterisation of the Element Depth Distribution

The depth distribution of the Fe and Si, and Ge atoms were determined with Rutherford backscattering spectroscopy using helium ions, He^+^, at 1.504 MeV and a scattering angle of 160° relative to the beam’s propagation direction. Two structural models of the layer stack were used to simulate the experimental spectra. One is a trilayer structure with abrupt interfaces (trilayer model). The second model consisted of 10 layers intended to account for the atomic diffusion and formation of intermediate buffer layers (gradient model). Figure 5a represents experimental and simulated spectra. It can be seen that each model describes the spectra (Figure 5a residual) well. However, the gradient model results in a minor discrepancy to the experimental data in the channel regions of 275–310, which corresponds to the bottom of the lower Fe_3+*x*_Si_1−*x*_ and Ge layers. This fact allows us to conclude (Figure 5c) on the presence of a Ge-enriched layer between the Ge and Fe_3+*x*_Si_1−*x*_ layers and diffusion of the Fe into the Si substrate. The Ge content may reach ~38 at.% in the 2 nm interface region of the Ge layer with lower Fe_3+*x*_Si_1−*x*_. Germanium is incorporated into the upper layer of Fe_3+*x*_Si_1−*x*_ in less concentration, up to 18 at.% in 2 nm adjacent to the Ge intermediate layer (Figure 5c). Comparing interatomic distance misfits for the [−110] direction (Figure 5b) calculated based on the chemical composition distribution, one can see that the average misfit between both models differs by one per cent, which indicates that a more extended exposition to the elevated temperature of the Fe_3+*x*_Si_1−*x*_/Ge/Fe_3+*x*_Si_1−*x*_/Si(111) heterostructure may relieve the epitaxial stress. However, there must be a balance in the temperature-deposition rate and deposition time to obtain an all-epitaxial layer heterostructure with desired thicknesses. It may be seen that incorporation of Ge atoms in the lower Fe_3+*x*_Si_1−*x*_ silicide layer along with the Si atom diffusion from the substrate may relax the epitaxial stress of this layer with the silicon substrate, but conversely results in its increase with the upper germanium layer (Figure 5b). In turn, the germanium layer tends to reduce the interface area with the lower Ge-enriched Fe_3+*x*_Si_1−*x*_ silicide through the formation of 3D islands (Figure 1f).

It is worth noting that the observed asymmetry of peaks on spectrum corresponding to iron and germanium are not accounted for by the layer thickness variation of the trilayer model (Figure 6). The asymmetry of the peaks was assessed with the bigaussian function. The experimental values of widths of biguassian function defined in units of RBS channel are w_1(Fe)_ = 3.3, w_2(Fe)_ = 2.8, and w_1(Ge)_ = 3.35, w_2(Ge)_ = 3.08. While the trilayer model shows the closest value w_1(Fe)_ = 2.83, w_2(Fe)_ = 3.09, and w_1(Ge)_ = 3.23, w_2(Ge)_ = 3.09. The gradient model is better suited to describe the experimental data fitted with bigaussian function widths w_1,2(Fe, Ge)_ equal to 3.28, 2.79, 3.28, and 2.8, respectively (Figure 6). Since the porosity of sample #7 reaches 17.3% (details are given below), the porosity was included in the simulation as a variable in the upper layers in the gradient and trilayer models. Simulations with porosity are excluded from the discussion since it does not change the simulated spectra noticeably.

#### 3.1.4. Surface Morphology and Dislocation Characteristics

We also characterized the topography by non-destructive atomic force microscopy (AFM). The distribution of roughness data is given in Table 3. Sample 7 is expected to have higher roughness due to the thicker Ge layer, the upper layer’s polycrystalline nature, and more significant misfits of Ge with the underlying Fe_3+*x*_Si_1−*x*_ layer [20,41]. The value of residual stress discussed above and the measured RMS values can be compared with the ones reported elsewhere [42]. Areas of size 20 × 20 µm^2^ and 2 × 2 µm^2^ were examined, statistical parameters for each sample were calculated in three different areas, and the mean value was calculated. Scanning areas with a size of 20 × 20 μm^2^ showed that the surface of the films is smooth, homogeneous, and does not have pronounced features. Figure 7a shows a typical topography of sample #7. When scanning a 2 × 2 µm^2^ area with a higher resolution (~7.8 nm per pixel), it is possible to distinguish nano-sized depressions (“pits”) on the film surface. Their surface density is different for samples #6 and #7, that is 17.3% and 43.1%, respectively. A significant difference in the thin surface morphology indicates different formation mechanisms for the samples discussed. Variation of the thermal history of the samples results in different levels of incorporation of Ge atoms into Fe_3+*x*_Si_1−*x*_ layers and causes variation of residual stress. The observable surface morphological characteristics are to be further analysed.

The typical pore size derived from the autocorrelated distribution function is 22.5 nm (Figure 8a), which is the same for both samples while the porosity of sample #7 is 2.5 times larger. Moreover, the pore size distribution is almost identical with the three most prominent pore sizes, which are 4.5, 18, and 25 nm (Figure 8c). Such a phenomenon refers to a different dominant mechanism of pore formation in epitaxial and polycrystalline upper Fe_3+*x*_Si_1−*x*_ layers. Under the condition of the same pore size, a sample with higher porosity would indicate less wettability of the surface, i.e., higher interface energy and lattice misfits. The forming layer tends to develop side facets and increase in bulk volume of 3D islands. The condition of the same amount of Fe–Si deposited should result in a different thickness of the upper Fe_3+*x*_Si_1−*x*_ layer. Otherwise, the Fe–Si are redistributed over the rough interface between the upper Fe_3+*x*_Si_1−*x*_ and intermediate Ge layers so that the thickness determined in a 2D projection measured with TEM remains the same in both samples (Figure 2).

Autocorrelated radial distribution function (RDF) for higher radii shows the prominent sharp peaks but with low probability (~0.01) (Figure 8b). It is also noticeable that the peaks in the 50–300 nm region between values are similar for both samples. The discrepancy in positions at higher radii may refer to the different spatial textures of overlapped pores. For the case of the epitaxial layer, the pore spatial distribution may correlate with the hexagonal lattice of the dislocation (Figure 9). The 2D dislocation lattice presented in Figure 9b was constructed by overlapping two lattices of stoichiometric Fe_3_Si and silicon with the experimental OR [43]. Figure 9b also shows the distribution of near coincidence sites on the interface of stoichiometric iron silicide [33,44]. The structural motif of the spatial distribution of the dislocation borders can be easily observed on the dark field images by the TEM technique in over-focus mode measured in plan view for the one 40 nm thick Fe_3+*x*_Si_1−*x*_ layer grown on Si(111) with the same procedure described above. The fast Fourier transform (Figure 9c) and distance between honeycomb borders (Figure 9d,f) observed correlate well with the expected interatomic misfit value of −4.8% along <110> (Figure 9e). Thus, the interface energy is supposed to increase at the awaited dislocation site, so a pore is predominantly located at such position until the amount of the deposited material allows growing a noncontinuous film under a given energy landscape. Since the epitaxial islands mostly form due to the Terrace–Ledge–Kink (TLK) mechanism [45] on the vicinal silicon surface (111), the density of the material can also be regulated by the average width of silicon terraces and show correlation with the average terrace width value. For the case of the polycrystalline film, the dislocation lattice should not affect the pore formation.

Here, we attempted to fit the autocorrelated RDF function with two simple models of hexagonal and rectangular lattice to indicate the mechanism of pore formation in the samples discussed. The hexagonal lattice corresponds to the dislocation sites for Fe_3+*x*_Si_1−*x*_(111)||Ge(111) along <112> direction (Figure 9b). The square lattice corresponds to the terrace model. We used one hexagonal lattice and several rectangular lattices (1 × 1, 2 × 1, and 4 × 1). The RDF function is described with the sum of three Gaussian functions corresponding to the pore size distribution (Figure 8c). The fitted parameters are *x*, *q*, and *w*, where *x* refers to the scale, i.e., distances between pores, *w* is the parameter regulating the width of the Gaussian function, and *q* is the fraction of the hexagonal model. The *1-q* value is a fraction of the rectangular pore lattice. The uniformly distributed array of random values of the fitted parameters with size 6 × 10^7^ was used to find the standard deviation function. The figure presents the 2D kernel density of the STD value and functional parameter of *x* and *q* for both samples. The fitted *w* is close to the experimental values determined from the pore size distribution (Figure 8c).

Under the suggestion, the mechanism of distribution of the pores in all-epitaxial Fe_3+*x*_Si_1−*x*_/Ge/Fe_3+*x*_Si_1−*x*_/Si(111) trilayer (sample #6) discussed above are mainly governed by the hexagonal dislocation lattice (Figure 10), and in the polycrystalline layer of Fe_3+*x*_Si_1−*x*_ it should be affected by the TLK growth mode; then, one can determine the average dislocation distance along [11−2] or average terrace width of Si(111) surface. The fitting procedure of the autocorrelated RDF (Figure 8b) reveals that sample #6 is better described with the hexagonal lattice and vice versa sample #7 finds the best solution for almost pure terrace model (Figure 10) with average terrace width close to 17.7 nm. This value is in good agreement with the Si(111) miscut of the silicon wafer used in our experiments (±1° as stated by the producer) [46,47].

Thus, prolonged exposition of the Fe_3+*x*_Si_1−*x*_ bottom layer at the higher temperature of 300 °C causes increased atomic diffusion on the interfaces resulting in higher lattice misfits of Ge/Fe_3+*x*_Si_1−*x*_ due to the incorporation of Ge atoms into the Fe_3+*x*_Si_1−*x*_ bottom layer. The film develops a three-dimensional surface to lower the total free energy in heteroepitaxial thin film systems with biaxial stress. As a result, the growing Ge layer in the surface valley should show higher residual stress than at a peak, so incoming atoms preferably attach to the peak area, increasing the surface roughness. According to our estimation, the misfit value in the case of all-epitaxial trilayer structure (sample #6) is |0.42|%, which is close to the estimation of the misfit value based on the nominal composition of the layer with abrupt interfaces (Figure 5b). The misfit value for the germanium layer (sample #6) is expected to be also close to 0.42% while for the sample #7 with higher atomic diffusion, the misfits for two upper layers are close −1% and 2% (Figure 5b). The iron-rich composition of the silicide without termination with ultra-thin silicon layer [11] also impacts increasing the interface energy. Altogether, this lowers the total energy through the developing surface and the transition to the polycrystalline growth while the growing layer remains textured with certain crystallites preserving higher residual stress than in the epitaxial layer. The texturing of the polycrystalline Fe_3+*x*_Si_1−*x*_ layer is expected to be due to the faceting of the monocrystalline Ge 3D islands.

### 3.2. Magnetic Properties

Differences in microstructure, degree of crystallinity, and residual stress should significantly affect magnetic properties [48,49], such as magnetization, coercive force, and anisotropy. Indeed, the saturation magnetization (Figure 11) of sample #6 with 4 nm thick Ge (“Ge 4 nm”) is 40% higher than that of sample # 7. The coercive force *H_C_* for this sample is almost three times lower (0.145 mT), while for sample #7 with 7 nm thick Ge (“Ge 7 nm”), it is 0.403 mT. In addition, sample #7 “Ge 7 nm” demonstrates some features of magnetization reversal indicated by arrows in Figure 11, which may be associated with the presence of additional ferromagnetic phases, disordered A2 Fe–Ge–Si alloys formed near the interfaces, non-stoichiometric composition, and imperfection of the crystal structure of the upper Fe_3+*x*_Si_1−*x*_ layer. Moreover, lower Fe_3+*x*_Si_1−*x*_ layers can also be different due to a two-times more extended exposition at elevated temperature of “Ge 7 nm” sample that leads more complex interface structure and composition on the lower Ge/Fe_3+*x*_Si_1−*x*_ and Fe_3+*x*_Si_1−*x*_/Si boundaries resulting to decreasing of saturation magnetisation M_S_ other loop’s features.

To determine the magnetic anisotropy, we measured the angular dependence of the ferromagnetic resonance (FMR) spectra both in the plane and perpendicular to the plane at a temperature of 300 K. The spectrum (Figure 12a) for sample #6 “Ge 4 nm” shows two absorption lines, which, given the RHEED and TEM data, most likely correspond to FMR of the upper and lower Fe_3+*x*_Si_1−*x*_ films. For sample #7 “Ge 7 nm” (Figure 12a), the second line is less pronounced and consists of two, potentially three, absorption lines. We analysed the angular dependence of all five lines (Figure 12b–f). The structure has an easy plane magnetization due to shape anisotropy. In addition, polar dependences reveal no magnetic coupling between upper and lower films of iron silicide. In particular, crossing of lines was observed during sweeping of out-of-plane angle. It occurs due to films having different M_S_ and corresponding shape anisotropy filed 4π·M_S_. Secondly, the angular in-plane dependence of the resonance field is characteristically different for all resonances (Figure 12).

Using the conventional analysis [50,51], we calculate the different contributions to the magnetic anisotropy (Table 4): saturation magnetization (M_S_), anisotropy field (H_K_), and anisotropy angle (α_K_) measured relative to the direction of uniaxial anisotropy, that is, the easy magnetization axis of the 1st line for each of the samples. It is worth noting that the M_S_ value of the 1st line derived from the resonance field is comparable to high-quality epitaxial Fe_3+*x*_Si_1−*x*_ films grown separately on different substrates (see for example [35,40,52]). For the 2nd line, M_S_ is smaller due to the deterioration of the crystalline quality of the upper film. The nature of the third line in sample #7 “Ge 7 nm” is associated with the imperfection of the upper Fe_3+*x*_Si_1−*x*_ layer and the presence of Fe–Si–Ge alloy at the interfaces. The uniaxial in-plane anisotropy is the dominating contribution. This behaviour was observed earlier for (111) Fe_3+*x*_Si_1−*x*_ films [53] and can be related to the surface of the Si substrate and Si(111)/Fe_3+*x*_/Si_1−*x*_(111) interface effects. Another reason may be associated with the features of an oblique deposition, which were observed earlier in Fe and other iron silicide films [54]. Nonetheless, we observe uniaxial anisotropy for sample #7 “Ge 7 nm”. If oblique deposition is the reason for the uniaxial anisotropy, it should be the same for all grown films. From the variation of the symmetry axis between films we conclude that the anisotropy directions are due to crystalline features in the layers not produced by oblique incidence of the atoms during growth.

There is also a noticeable 6th order contribution Hk_6_ (Table 4) along with the cubic Hk_4_ (4th order) symmetry axis [52,55,56]. For example, for lines presented on Figure 12b we found that Hk_6_ is greater than Hk_4_. Moreover, Hk_6_ is only 2.7 times lower than uniaxial Hk_2_ contribution. The reason for this is that both the upper and lower Fe_3+*x*_Si_1−*x*_ films have an epitaxial relation with the Si (111) || Fe_3+*x*_Si_1−*x*_ (111), which means that we observe the anisotropy of the (111) Fe_3+*x*_Si_1−*x*_ crystal, which has a six-fold crystal symmetry (Figure 2c).

### 3.3. Transport Properties

In addition to certain magnetic properties, three-layer FM/SC/FM structures must also have specific transport properties. In particular, the SC layer must exhibit semiconducting transport properties, which is important for controlling spin transport using an electric field or, for example, optical irradiation. Using etching in an HF: HNO_3_: H_2_O = 1: 2: 400 solution, we prepared samples with contacts made from the Fe_3+*x*_Si_1−*x*_ film on the Ge surface (inset in Figure 13) to measure the temperature dependences of the Ge resistance and compare it with the films of different thicknesses. The distance between Fe_3+*x*_Si_1−*x*_ contacts was 500 µm. Note that the contact remains ohmic up to 5 K, i.e., the *I–V* curves are linear over the entire temperature range for both samples. The resistance R of sample #6 “Ge 4 nm” increases nonlinearly with decreasing temperature rising by about a factor of 3 at 5 K compared with R at 300 K (left panel in Figure 13). At the same time, sample #7 “Ge 7 nm” demonstrates a completely different behaviour; upon cooling, its resistance first decreases monotonically, reaching a minimum at 55 K, and then increases (right panel in Figure 13). Moreover, the relative changes are minimal. The ratio of the minimum resistance to the room temperature resistance R_min_/R_300_ is 0.95, which is a 5% change. Over the entire temperature range, resistance changes for only 2.5% (R_5_/R_300_ = 0.975).

It can be seen from Figure 13 that the temperature dependence of the resistance for sample #6 “Ge 4 nm” is due to the conductivity of the thermoactivation type. However, fitting in Arrhenius coordinates (ln (R) vs. 1/T) does not give a good linear fit. The best linearization is obtained using R = R_0_ exp(T_0_/T^1/4^) (Figure 14), which suggests that the hopping type of conduction with variable hopping length (VRH) [57] prevails over the thermal delocalization of carriers and their transfer to the conduction band of germanium. The calculated parameter T_0_ is 65 K, which is typical for “inhomogeneous” materials [58]. The dominance of the VRH mechanism is most likely due to the small thickness of the Ge interlayer and, accordingly, the high density of defects, that may indicate a partial island-like character of the film. Measured magnetoresistance curves R(H) at temperature 4 K in parallel and perpendicular magnetic field (not shown here) found no differences that is an additional argument for the 3-dimensional VRH transport mechanism.

The resistance of sample #7 “Ge 7 nm” resembles a degenerate semiconductor. Considering slight changes with temperature (R_5_/R_300_ = 0.975), we believe that a decrease in resistance in the temperature range from 300 K to 55 K is associated with an increase in the mobility of charge carriers due to a decrease in electron-phonon scattering. When analysing the low-temperature part of the curve, we tried various models such as the thermally activated behaviour (ln(R)~1/T), thermoactivated tunnelling between nearest grains in a granular system (ln(R)~1/T^1/2^) [59], and the VRH (ln(R)~1/T^1/4^) behaviour for fitting. However, none of these models fit the experimental data well. It can be assumed that an increase in resistance below 55 K might be caused by scattering of magnetic impurities, i.e., Kondo scattering [60]. As it was shown, Fe impurities are likely to be present at the Ge/Fe_3+*x*_Si_1−*x*_ interfaces. Another reason may be the manifestation of quantum corrections to the conductivity at low temperatures [61].

## 4. Conclusions

Epitaxial Fe_3+*x*_Si_1−*x*_/Ge/Fe_3+*x*_Si_1−*x*_ trilayers with 4 and 7 nm Ge layer thickness were grown. At a higher temperature of 300 °C during the formation of the Ge layer, we found a larger lattice misfit at the Ge/Fe_3+*x*_Si_1−*x*_ interface and the development of a rough interface due to the incorporation of Ge atoms into the Fe_3+*x*_Si_1−*x*_ bottom layer. We demonstrate the epitaxial growth of an iron-rich Fe_3+*x*_Si_1−*x*_ upper layer on the germanium layer with thickness of 4 nm. With increasing Ge thickness, the upper Fe_3+*x*_Si_1−*x*_ layer becomes polycrystalline and with a rougher surface.

The ferromagnetic resonance study revealed magnetic crystal anisotropy with a six-fold symmetry typical for the (111) plane of Fe_3+*x*_Si_1−*x*_. This fact indicates the probable epitaxial ratio Fe_3+*x*_Si_1−*x*_(111)[0−11] || Ge(111)[1−10] || Fe_3+*x*_Si_1−*x*_(111)[0−11] || Si(111)[1−10]. The resistance of the Ge layer measured on specially prepared structures demonstrates an increase with decreasing temperature, reflecting its semiconductor nature. With an increase in the Ge thickness to 7 nm, the transport properties become similar to a degenerate semiconductor due to intermixing iron, germanium, and silicon atoms in disordered interfaces.

Our work showed that in the iron-rich Fe_3+*x*_Si_1−*x*_/Ge/Fe_3+*x*_Si_1−*x*_ system, a high crystalline perfection of individual layers can be obtained while maintaining the semiconducting properties of the Ge layer. At the same time, an increase in the semiconductor layer thickness from 4 nm to 7 nm leads to a significant change in the magnetic properties of the upper ferromagnetic layer, yielding the opportunity to vary its magnetization by controlling the Ge thickness. Additionally, we discussed the limits of the all-epitaxial formation of the iron-rich Fe_3+*x*_Si_1−*x*_/Ge/Fe_3+*x*_Si_1−*x*_ heterostructures and its relation to the structural, magnetic, and transport properties.

## Figures and Tables

**Figure 1 nanomaterials-12-00131-f001:**
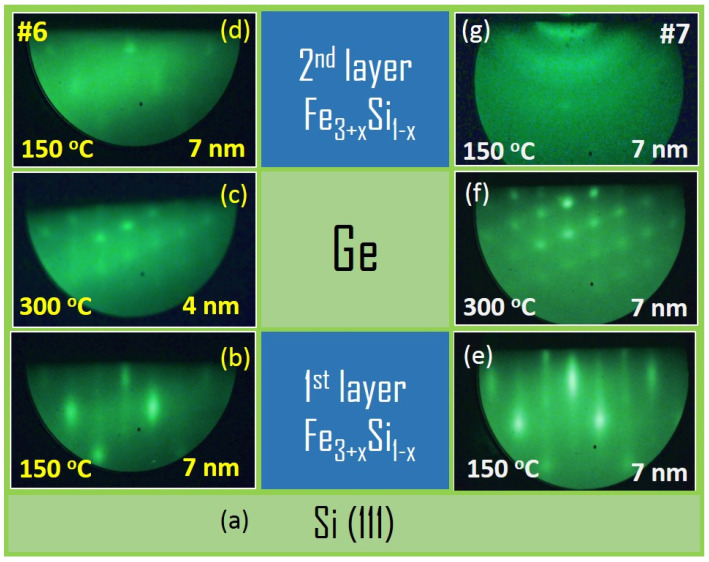
(**a**) Middle panel shows the schematics of the three-layer structure Fe_3+*x*_Si_1−*x*_/Ge/Fe_3+*x*_Si_1−*x*_/Si(111). The panels on the left and right show the respective RHEED patterns obtained after deposition of the respective layer for samples #6 and #7: (**b**)—1st Fe_3+*x*_Si_1−*x*_, (**c**)—Ge 4 nm, (**d**)—2nd Fe_3+*x*_Si_1−*x*_ and (**e**)—1st Fe_3+*x*_Si_1−*x*_, (**f**)—Ge 7 nm, and (**g**)—2nd Fe_3+*x*_Si_1−*x*_, respectively.

**Figure 2 nanomaterials-12-00131-f002:**
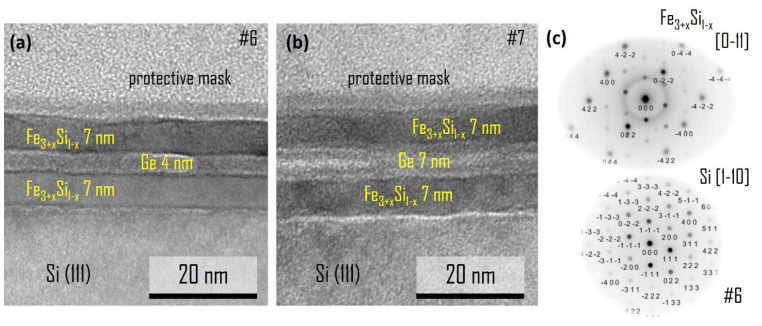
TEM images of cross section of Fe_3+*x*_Si_1−*x*_/Ge/Fe_3+*x*_Si_1−*x*_/Si(111) samples #6 (**a**) and #7 (**b**). Unit cell and crystallographic plane with iron atoms of the Fe_3+*x*_Si_1−*x*_ epilayer in the (111) film plane (**c**).

**Figure 3 nanomaterials-12-00131-f003:**
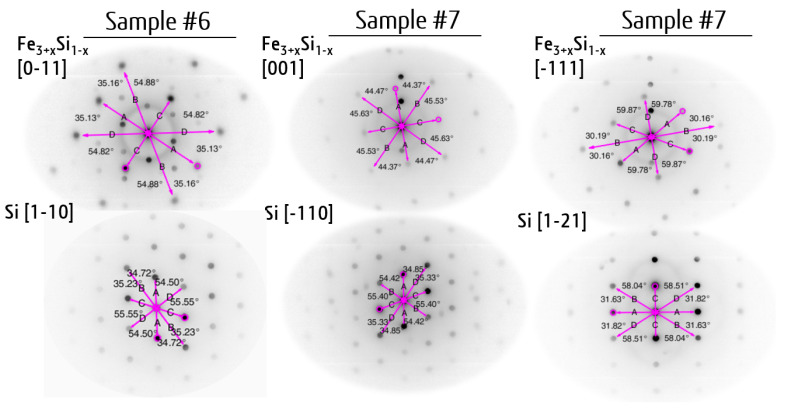
TEM electron diffraction pattern of cross-section of Fe_3+*x*_Si_1−*x*_/Ge/Fe_3+*x*_Si_1−*x*_/Si(111) samples #6 and #7 along different projections (zone axes). The angles of reciprocal lattice vectors are given for each phase.

**Figure 4 nanomaterials-12-00131-f004:**
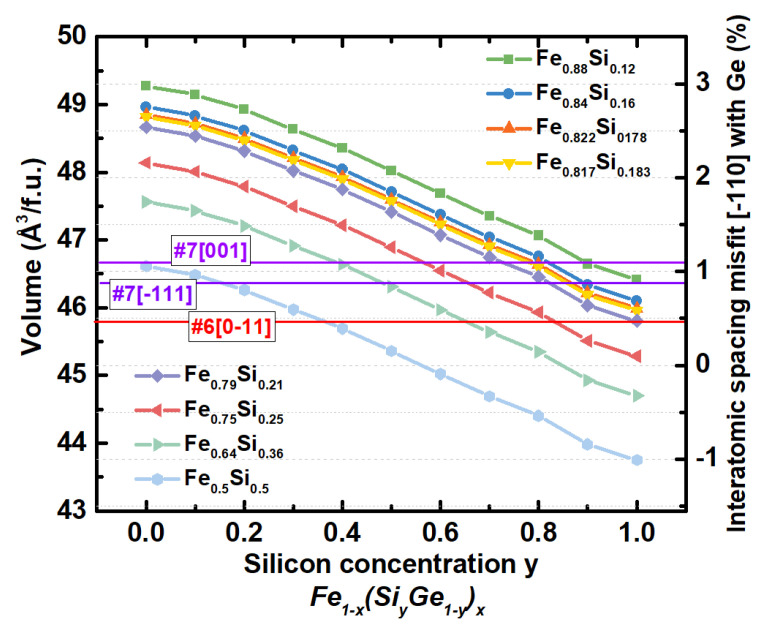
Dependence of unit cell volume per functional unit of ternary Fe–Ge–Si alloys.

**Figure 5 nanomaterials-12-00131-f005:**
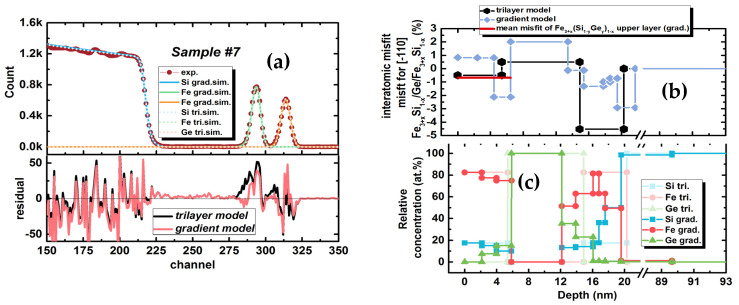
RBS spectra for Fe_3+*x*_Si_1−*x*_/Ge/Fe_3+*x*_Si_1−*x*_/Si(111) (sample #7) (**a**), misfits of each layer calculated based on the different fitting model of RBS spectra (**b**), a profile of relative chemical element concentration (at.%) derived from the RBS measurements for gradient and trilayer model (**c**).

**Figure 6 nanomaterials-12-00131-f006:**
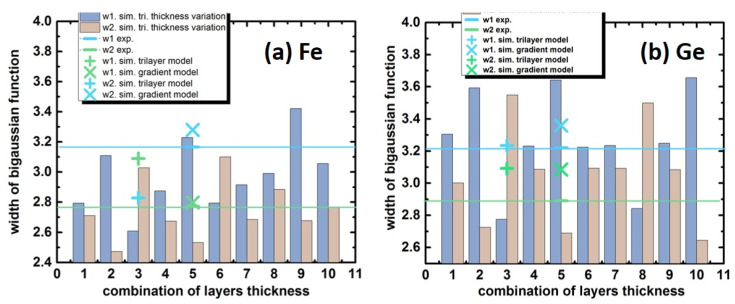
Analysis of asymmetry of the peaks of the RBS spectra for Fe_3+*x*_Si_1−*x*_/Ge/Fe_3+*x*_Si_1−*x*_/Si(111) (sample #7) with bigaussian function; blue and green lines refer to the experimental values of peak asymmetry observed for (**a**) Fe and (**b**) Ge; green marks corresponded to trilayer or gradient model fits as discussed; and bars indicate the asymmetry values for the different combinations of thickness in the trilayer model.

**Figure 7 nanomaterials-12-00131-f007:**
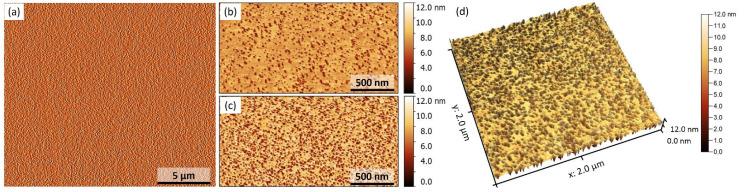
AFM images of the surface of Fe_3+*x*_Si_1−*x*_/Ge/Fe_3+*x*_Si_1−*x*_/Si(111) films. (**a**) A typical depiction of surface topology for sample #7; (**b**) for sample #6; (**c**) for sample #7; and (**d**) 3-d surface topology view for sample #6.

**Figure 8 nanomaterials-12-00131-f008:**
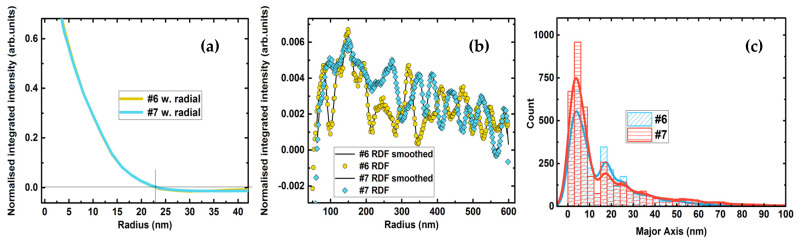
The radial average of the autocorrelation of pores from AFM images of the surface of Fe_3+*x*_Si_1−*x*_/Ge/Fe_3+*x*_Si_1−*x*_/Si(111) heterostructure. (**a**) The average pore size is close to 23 nm for both samples; (**b**) autocorrelated RDF for the samples #6 and #7 beyond the average pore size; and (**c**) distribution of pore size fitted with ellipses for the sample discussed.

**Figure 9 nanomaterials-12-00131-f009:**
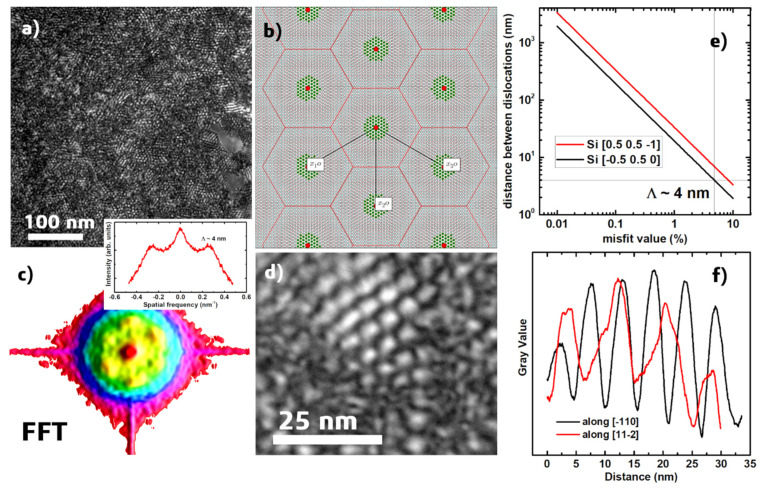
(**a**) A TEM plan-view image of one layer of Fe_3+*x*_Si_1−*x*_ grown on Si(111) (over-focus mode); (**b**) 2D O-lattice formed by overlapping two lattices, lattice points of which are represented by small, red circles (Si) and cyan circles (Fe_3_Si), respectively. Each O-point (large circle) is at the centre of an O-cell (O-cell walls—solid lines), and near coincidence sites are depicted with green-filled circles; (**c**) FFT image in a 3D perspective view of Figure 9a. The inset depicts a line cut over the FFT image. (**d**) Magnified view of the structural motif of the dislocation lattice; (**e**) dependence of distance between dislocation along [−110] and [11−2] directions on misfit value for silicon; and (**f**) intensity distribution along two directions of the structural motif depicted on Figure 9d.

**Figure 10 nanomaterials-12-00131-f010:**
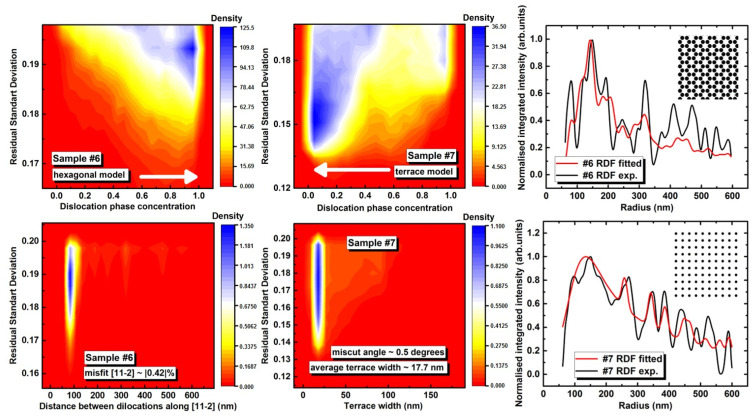
Dependence of 2D kernel density of residual standard deviation on random uniform distribution of sets fitting parameters of the autocorrelated RDF for both samples: (**upper left**) weight of pore distribution model (hexagonal or 1 × 1 square lattice or terrace), (**lower left**) distance between the dislocation along [11−2], sample #6; (**middle-upper**) weight of pore distribution model (hexagonal or terrace), (**middle-****low****er**) distance between average terrace width, sample #7; (**upper right**) the autocorrelated RDF fitted for the sample #7, (**right lower**), the autocorrelated RDF fitted for the sample #6; insets show the pore distribution model.

**Figure 11 nanomaterials-12-00131-f011:**
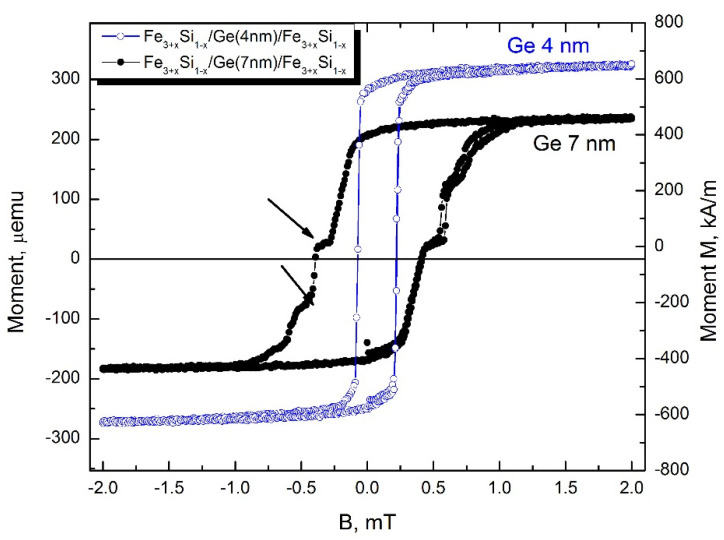
In-plane magnetization reversal of Fe_3+*x*_Si_1−*x*_/Ge(4 nm)/Fe_3+*x*_Si_1−*x*_ and Fe_3+*x*_Si_1−*x*_/Ge(8 nm)/Fe_3+*x*_Si_1−*x*_ structures at 300 K.

**Figure 12 nanomaterials-12-00131-f012:**
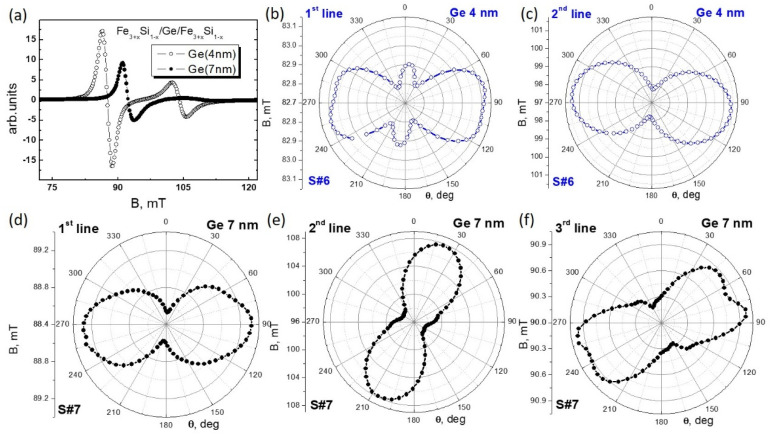
(**a**) FMR spectra of Fe_3+*x*_Si_1−*x*_/Ge(4 nm)/Fe_3+*x*_Si_1−*x*_ (**a**) and Fe_3+*x*_Si_1−*x*_/Ge(7 nm)/ Fe_3+*x*_Si_1−*x*_ structures. Polar plots of angular dependences of 1st (**b**) and 2nd (**c**) lines of Fe_3+*x*_Si_1−*x*_/Ge(4 nm)/Fe_3+*x*_Si_1−*x*_ structure and 1st (**d**), 2nd (**e**), and 3rd (**f**) lines of Fe_3+*x*_Si_1−*x*_/Ge(7 nm)/Fe_3+*x*_Si_1−*x*_ structure.

**Figure 13 nanomaterials-12-00131-f013:**
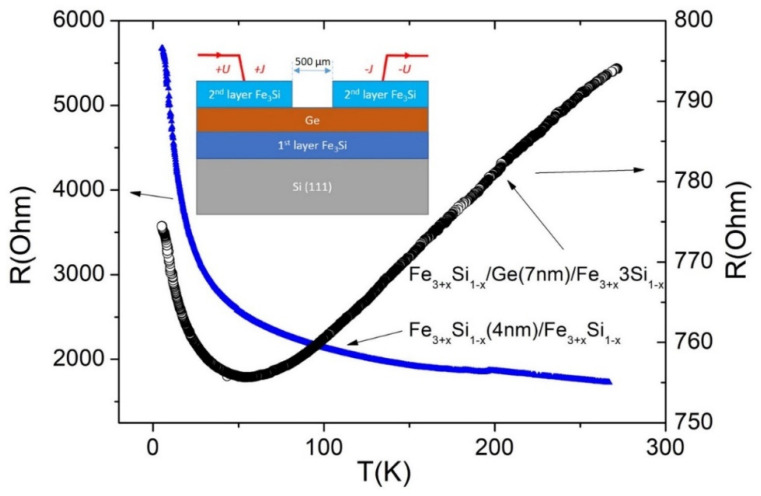
Temperature dependences of resistance of Fe_3+*x*_Si_1−*x*_/Ge(4 nm)/Fe_3+*x*_Si_1−*x*_ and Fe_3+*x*_Si_1−*x*_/Ge(7 nm)/Fe_3+*x*_Si_1−*x*_ etched structures.

**Figure 14 nanomaterials-12-00131-f014:**
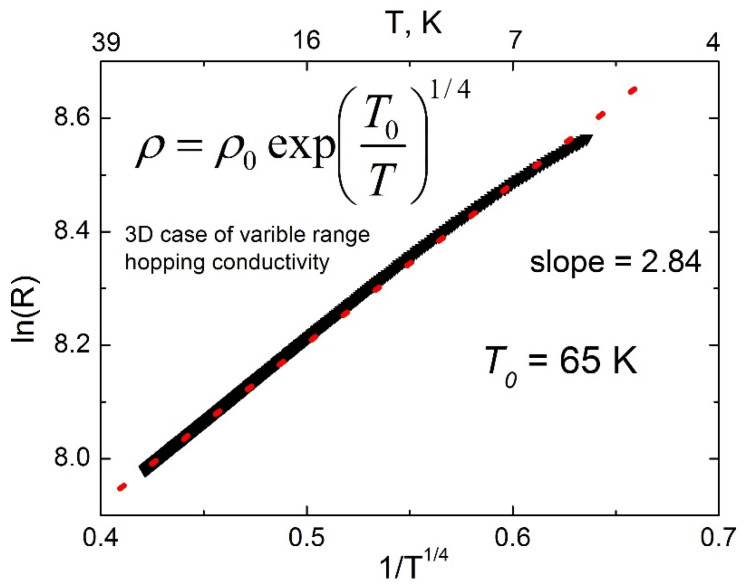
Fitting of temperature dependence of resistance of Fe_3+*x*_Si_1−*x*_/Ge(4 nm)/Fe_3+*x*_Si_1−*x*_ etched structure.

**Table 1 nanomaterials-12-00131-t001:** Comparison of the true values of lattice distortions of a cubic lattice with the numerical solution found for two types of calculations δ_a_ = δ_b_ ≠ δ_c_, δ_α_ = δ_β_ ≠ δ_γ_. The sign of ± indicates the standard deviation from the average value of the 200 solutions with the minimal residual function observed.

	Residual	δ_a,b_, nm	δ_c_, nm	δ_α,β_, deg.	δ_γ_, deg.
True values of quantities		**−1.991 × 10^−3^**	**4.2908 × 10^−2^**	**−0.07512**	**0.12**
Solutions	2.13 × 10^−10^ ± 6.44 × 10^−11^	−1.991 × 10^−3^ ± 5.99 × 10^−8^	4.279 × 10^−2^ ± 1.69 × 10^−5^	−0.07629 ± 0.0017	0.1212 ± 0.0017
	9.34 × 10^−10^ ± 2.48 × 10^−10^	−1.991 × 10^−3^ ± 2.31 × 10^−7^	6.205 × 10^−2^ ± 7.46 × 10^−5^	0.114 ± 0.0074	−0.0668 ± 0.0072

**Table 2 nanomaterials-12-00131-t002:** Solutions for lattice distortions for the Fe_3+*x*_Si_1−*x*_ epilayer of sample #6 and two crystallites of sample #7. Configurations of crystal lattice distortions are given in the table. The sign of ± indicates the standard deviation from the average value of the 200 solutions with the minimal residual function observed.

		Residual	δ_a_, nm	δ_b_, nm	δ_c_, nm	δ_α_, deg.	δ_β_, deg.	δ_γ_, deg.
**Sample #6 cubic**								
**Solutions found**	δ_a_ ≠ δ_b_ = δ_c_, δ_α_ = δ_β_ = δ_γ_	1.11 × 10^−6^ ± 1.31 × 10^−16^ 1.11 × 10^−6^ ± 6.11 × 10^−12^ 1.18 × 10^−6^ ± 4.67 × 10^−11^	−1.199 × 10^−3^ ± 2.55 × 10^−11^ −1.200 × 10^−3^ ± 3.66 × 10^−7^ −1.148 × 10^−3^ ± 1.34 × 10^−6^	2.657 × 10^−3^ ± 1.29 × 10^−13^ 2.655 × 10^−3^ ± 6.14 × 10^−9^ 2.656 × 10^−3^ ± 3.31 × 10^−8^		−0.0165 ± 2.15 × 10^−9^ −0.0173 ± 1.17 × 10^−5^ 8.181 × 10^−4^ ± 5.72 × 10^−6^		
	δ_a_ ≠ δ_b_ = δ_c_, δ_α_ = 0; δ_β_ ≠ δ_γ_	1.50 × 10^−8^ ± 7.49 × 10^−14^	−1.368 × 10^−3^ ± 1.82 × 10^−10^	1.00 × 10^−3^ ± 3.33 × 10^−10^		0	0.0163 ± 4.07 × 10^−6^	0.0208 ± 4.06 × 10^−6^
		1.50 × 10^−8^ ± 1.25 × 10^−13^	−1.368 × 10^−3^ ± 2.75 × 10^−9^	1.00 × 10^−3^ ± 3.49 × 10^−9^		0	−0.0442 ± 2.26 × 10^−6^	0.0814 ± 2.17 × 10^−6^
		1.50 × 10^−8^ ± 3.62 × 10^−14^	−1.368 × 10^−3^ ± 4.39 × 10^−11^	1.00 × 10^−3^ ± 7.15 × 10^−11^		0	0.0605 ± 2.87 × 10^−6^	−0.0234 ± 2.88 × 10^−6^
		1.50 × 10^−8^ ± 8.83 × 10^−14^	−1.368 × 10^−3^ ± 9.75 × 10^−10^	1.00 × 10^−3^ ± 3.00 × 10^−9^		0	0.0794 ± 1.00 × 10^−5^	−0.0422 ± 1.01 × 10^−5^
		1.50 × 10^−8^ ± 8.22 × 10^−14^	−1.368 × 10^−3^ ± 1.22 × 10^−10^	1.00 × 10^−3^ ± 1.35 × 10^−10^		0	0.0255 ± 8.56 × 10^−6^	0.0627 ± 8.57 × 10^−6^
	δ_a_ ≠ δ_b_ = δ_c_, δ_α_ ≠ δ_β_ = −δ_γ_	6.49 × 10^−7^ ± 4.54 × 10^−17^ 3.54 × 10^−7^ ± 8.57 × 10^−19^ 3.54 × 10^−7^ ± 4.98 × 10^−18^ 3.54 × 10^−7^ ± 5.47 × 10^−19^	−1.341 × 10^−3^ ± 4.77 × 10^−10^ −1.368 × 10^−3^ ± 1.20 × 10^−11^ −1.368 × 10^−3^ ± 5.72 × 10^−10^ −1.368 × 10^−3^ ± 2.25 × 10^−12^	−2.27 × 10^−6^ ± 1.45 × 10^−14^ 1.20 × 10^−3^ ± 1.05 × 10^−7^ 5.95 × 10^−^^4^ ± 2.86 × 10^−7^ 1.44 × 10^−3^ ± 3.47 × 10^−8^		−0.175 ± 2.71 × 10^−12^ 0.032 ± 2.12 × 10^−5^ −0.085 ± 5.80 × 10^−5^ 0.085 ± 6.96 × 10^−^^6^	0.0003 ± 6.71 × 10^−4^ 0.0002 ± 4.51 × 10^−6^ 0.0002 ± 2.52 × 10^−6^ 0.0002 ± 3.65 × 10^−7^	−0.0003 ± 6.71 × 10^−4^ −0.0002 ± 4.51 × 10^−6^ −0.0002 ± 2.52 × 10^−6^ −0.0002 ± 3.65 × 10^−7^
		3.54 × 10^−7^ ± 2.82 × 10^−20^ 3.54 × 10^−7^ ± 3.92 × 10^−20^	−1.367 × 10^−3^ ± 2.86 × 10^−10^ −1.367 × 10^−3^ ± 2.76 × 10^−10^	1.00 × 10^−3^± 3.79 × 10^−^^8^ 1.00 × 10^−3^± 4.07 × 10^−^^8^		0.001 ± 7.64 × 10^−^^6^ 0.001 ± 8.20 × 10^−^^6^	−0.087 ± 9.32 × 10^−^^6^ 0.087 ± 9.14 × 10^−^^6^	0.087 ± 9.32 × 10^−^^6^ −0.087 ± 9.14 × 10^−^^6^
	δ_a_ ≠ δ_b_ = δ_c_, δ_α_ ≠ δ_β_ ≠ δ_γ_	1.07 × 10^−6^ ± 8.82 × 10^−^^12^ 1.50 × 10^−^^8^ ± 8.84 × 10^−15^ 1.50 × 10^−^^8^ ± 4.83 × 10^−15^ 1.50 × 10^−^^8^ ± 3.30 × 10^−15^	−1.341 × 10^−3^ ± 1.33 × 10^−^^7^ −1.368 × 10^−3^ ± 2.12 × 10^−9^ −1.368 × 10^−3^ ± 3.57 × 10^−9^ −1.368 × 10^−3^ ± 2.58 × 10^−9^	−7.99 × 10^−^^8^ ± 2.46 × 10^−9^ 1.10 × 10^−3^ ± 1.39 × 10^−^^6^ 6.69 × 10^−^^4^ ± 5.75 × 10^−^^7^ 1.37 × 10^−3^ ± 4.99 × 10^−^^7^		−0.145 ± 4.93 × 10^−^^7^ 0.018 ± 2.80 × 10^−^^4^ −0.070 ± 1.15 × 10^−^^4^ 0.071 ± 1.00 × 10^−^^4^	0.012 ± 3.10 × 10^−^^4^ 0.012 ± 1.78 × 10^−^^4^ 0.012 ± 8.63 × 10^−5^ 0.012 ± 1.014 × 10^−^^4^	0.024 ± 3.21 × 10^−^^4^ 0.025 ± 1.78 × 10^−^^4^ 0.026 ± 8.63 × 10^−5^ 0.025 ± 1.012 × 10^−^^4^
		1.50 × 10^−^^8^ ± 3.02 × 10^−^^14^	−1.368 × 10^−3^ ± 6.76 × 10^−9^	1.01 × 10^−3^ ± 3.04 × 10^−^^6^		−0.001 ± 6.13 × 10^−^^4^	−0.060 ± 1.87 × 10^−^^4^	0.097 ± 1.87 × 10^−^^4^
		1.50 × 10^−^^8^ ± 8.20 × 10^−15^	−1.368 × 10^−3^ ± 2.26 × 10^−9^	1.02 × 10^−3^ ± 1.12 × 10^−5^		0.001 ± 2.26 × 10^−^^4^	0.070 ± 1.22 × 10^−^^4^	−0.033 ± 1.23 × 10^−^^4^
		1.50 × 10^−^^8^ ± 8.97 × 10^−13^ 1.50 × 10^−^^8^ ± 1.57 × 10^−13^	−1.368 × 10^−3^ ± 1.77 × 10^−^^8^ −1.368 × 10^−3^ ± 1.10 × 10^−^^8^	9.94 × 10^−^^4^ ± 1.63 × 10^−5^ 9.97 × 10^−^^4^ ± 1.25 × 10^−5^		−0.005 ± 3.3 × 10^−3^ −0.004 ± 2.5 × 10^−3^	0.090 ± 6.70 × 10^−^^4^ −0.048 ± 4.38 × 10^−^^4^	−0.053 ± 6.69 × 10^−^^4^ 0.085 ± 4.37 × 10^−^^4^
	**Sample #6 hexagonal**							
	δ_a_ = δ_b_ ≠ δ_c_, δ_α_ ≠ δ_β_ ≠ δ_γ_ = 0	1.05 × 10^−6^ ± 7.75 × 10^−16^	0.0539 ± 1.88 × 10^−10^		−0.1532 ± 4.84 × 10^−11^	−4.74 × 10^−^^4^ ± 6.26 × 10^−11^	0.1439 ± 3.2 × 10^−11^	0
		1.50 × 10^−^^8^ ± 1.51 × 10^−^^12^	0.0539 ± 2.90 × 10^−^^8^		−0.1534 ± 7.37 × 10^−^^8^	0.1068 ± 4 × 10^−^^6^	0.1355 ± 2.25 × 10^−^^6^	0
		2.98 × 10^−6^ ± 2.13 × 10^−15^	0.0538 ± 1.7 × 10^−9^		−0.1529 ± 7.43 × 10^−11^	0.1217 ± 1.69 × 10^−10^	−2.32 × 10^−^^4^ ± 8.47 × 10^−11^	0
	δa = δ_b_ ≠ δ_c_, δ_α_ = −δ_β_ ≠ δ_γ_ = 0;	2.79 × 10^−6^ ± 2.09 × 10^−15^ 4.43 × 10^−6^ ± 1.98 × 10^−14^	0.0538 ± 8.61 × 10^−10^ 0.0538 ± 8.64 × 10^−9^		−0.1529 ± 1.00 × 10^−11^ −0.1529 ± 1.00 × 10^−10^	−0.1373 ± 2.43 × 10^−10^ 4.57 × 10^−4^ ± 1.86 × 10^−9^	0.1373 ± 2.43 × 10^−10^ −4.57 × 10^−4^ ± 1.86 × 10^−9^	0 0
	**Sample #7 cubic–zone axis [001]**							
	δ_a_ ≠ δ_b_, δ_c_ = 0, δ_α_ = δ_β_ = 0, ≠ δ_γ_	8.43 × 10^−^^7^ ± 4.45 × 10^−14^	5.45 × 10^−^^4^ ± 3.17 × 10^−11^	−4.55 × 10^−^^4^ ± 7.09 × 10^−11^	-	-	-	−1.177 ± 8.67 × 10^−^^8^
		9.64 × 10^−^^5^ ± 1.39 × 10^−^^7^ 8.09 × 10^−^^5^ ± 9.11 × 10^−^^8^	−1.29 × 10^−6^ ± 7.88 × 10^−^^7^ 3.16 × 10^−^^4^ ± 2.43 × 10^−^^7^	−1.83 × 10^−^^4^ ± 3.77 × 10^−^^7^ 1.64 × 10^−^^6^ ± 5.15 × 10^−^^7^	- -	- -	- -	−1.176 ± 9.71 × 10^−^^6^ −1.174 ± 4.34 × 10^−^^6^
	**Sample #7 cubic–zone axis [−111]**							
	δ_a_ ≠ δ_b_ ≠ δ_c_, δ_α_ ≠ δ_β_ ≠ δ_γ_	1.79 × 10^−^^5^ ± 6.77 × 10^−20^	−6.00 × 10^−^^3^ ± 2.60 × 10^−18^	2.13 × 10^−^^4^ ± 1.897 × 10^−19^	1.1 × 10^−^^3^ ± 8.67 × 10^−19^	1.87 × 10^−^^4^ ± 1.89 × 10^−19^	−0.9996 ± 4.22 × 10^−19^	4.9 × 10^−^^4^ ± 1.73 × 10^−18^
		1.79 × 10^−^^5^ ± 1.29 × 10^−10^	5.10 × 10^−^^3^ ± 5.79 × 10^−5^	4.61 × 10^−^^5^ ± 1.31 × 10^−^^4^	−1.1 × 10^−^^3^ ± 8.57 × 10^−^^4^	0.248 ± 0.0923	−0.1129 ± 0.0706	1.1023 ± 0.0545

**Table 3 nanomaterials-12-00131-t003:** Surface statistics parameters (over the entire scanning area).

	Scanning Area (μm)	Mean Value (nm)	RMS Roughness, Sq (nm)	Average Roughness Sa (nm)	Median, nm	Maximum Height Sz (nm)
#6	2 × 2	6.90	1.42	1.01	7.34	11.196
#6	20 × 20	1.198	0.284	0.228	1.22	2.44
#7	2 × 2	7.05	2.20	1.87	7.69	12.69
#7	20 × 20	4.58	1.12	0.91	4.61	9.21

**Table 4 nanomaterials-12-00131-t004:** Parameters of FMR lines and contributions of anisotropy of three-layer structures.

Ge			Anisotropy					
		Magnetization Saturation	Uniaxial		Four-Fold		Six-Fold	
	Sample	Ms, kA/m	Hk2, mT	αk2, deg.	Hk4, mT	Tk4, deg.	Hk6, mT	αk6, deg.
**4 nm**	1st line	1034.04	0.153	**0**	0.043	−41.29	0.058	−85.55
	2nd line	835.99	1.925	7.13	0.117	−81.27	0.058	−115.09
**7 nm**	1st line	955.13	0.366	**0**	0.046	−73.49	0.033	−52.04
	2nd line	814.07	5.613	−65.13	1.085	−25.42	0.210	−97.22
	3rd line	935.24	0.378	−23.39	0.0078	−76.28	0.043	−59.06

## Data Availability

Not applicable.

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
