# Peer review of "Asymmetric Interfaces in Epitaxial Off-Stoichiometric Fe3+xSi1−x/Ge/Fe3+xSi1−x Hybrid Structures: Effect on Magnetic and Electric Transport Properties"

_nanomaterials, 2021, doi:10.3390/nano12010131_

Round 1
Reviewer 1 Report
A detailed characterization of three-layer ferromagnetic/semiconductor/ferromagnetic structures has been reported in the manuscript. The thickness of the Ge central layer is of nanometer order, in particular, two thicknesses have been considered, namely 4 nm and 7 nm. As evinced by RHEED, TEM, and RBS characterization, the Ge thickness affects the crystal orientation of the component layers as well as the interdiffusion of Ge. The effects on electrical and magnetic properties were measured and discussed.
The manuscript is well written and both the structural and functional characterizations are clearly reported. The manuscript topic is interesting and timely, thus I recommend the publication on Nanomaterials after some minor points to be addressed. Please, find a list of them in the following.
Please, define FMR in the abstract.
Page 11: Please add the measure units of wi width values reported.
Page 12: Please define RDF
page 17: Please, add the distance between electrodes (see Fig. 13)
Fig. 14: in Fig. 14, the TM symbol is used instead of the T0 parameter cited in the text ( page 17).
Author Response
Reviewer 1
Comments and Suggestions for Authors
A detailed characterization of three-layer ferromagnetic/semiconductor/ferromagnetic structures has been reported in the manuscript. The thickness of the Ge central layer is of nanometer order, in particular, two thicknesses have been considered, namely 4 nm and 7 nm. As evinced by RHEED, TEM, and RBS characterization, the Ge thickness affects the crystal orientation of the component layers as well as the interdiffusion of Ge. The effects on electrical and magnetic properties were measured and discussed.
- The manuscript is well written and both the structural and functional characterizations are clearly reported. The manuscript topic is interesting and timely, thus I recommend the publication on Nanomaterials after some minor points to be addressed. Please, find a list of them in the following.
Thank you very match for your positive view of our work.
- Please, define FMR in the abstract.
“Ferromagnetic resonance (FMR)” was added in the abstract.
- Page 11: Please add the measure units of wiwidth values reported.
The units of wi width values were mentioned in the text and on the Figure 6.
“The experimental values of widths of bigaussian function defined in units of RBS channel…”
- Page 12: Please define RDF
Definition of the RDF abbreviation is now given in the text.
“Autocorrelated radial distribution function (RDF) for higher radii…”
- page 17: Please, add the distance between electrodes (see Fig. 13)
The distance between electrodes was 500 µm. It was added in text and denoted on scheme in Fig.13.
- 14: in Fig. 14, the TMsymbol is used instead of the T0 parameter cited in the text ( page 17).
In Fig.14 TM was changed for T0. p=p0 exp(T0/T)1/4
English language and style were slightly corrected and highlighted in yellow.
In the manuscript, all other changes are highlighted in green.

Reviewer 2 Report
The authors present the results of a very careful growth and characterization study of Fe3+xSi1-x/Ge/Fe3+xSi1-x trilayer structures. These have potential as vertical magnetotransport elements and are of particular interest because of the high spin polarization of the magnetic electrodes. From a practical point of view the growth of these structures is challenging due to lattice mismatch and the different required growth temperatures of the Fe3+xSi1-x and Ge layers. Two samples have been very carefully structurally, electronically and magnetically characterized and some important conclusions drawn. I believe that the paper should be suitable for publication in Nanomaterials once the following points have been addressed in a revised manuscript.
- The very comprehensive characterization of two samples with different Ge thicknesses has been described in the manuscript. We are given to understand that these effectively represent two different limits, one corresponding to epitaxial growth and one corresponding to polycrystalline growth. While it would be unreasonable to expect the same level of detailed characterization for a much larger number of samples, do the authors have data (even quite qualitative data) on samples with other Ge thicknesses which demonstrate that these two are truly illustrative of the two limits?
- The details of the structural analysis of the films are quite extensive and do not make very easy reading. The authors might want to consider moving some of the detail to an appendix and just retaining to key results in the main manuscript.
- The authors should state what protective layer was used when fabricating the TEM samples, was it Pt? It would be useful if they could reflect on how much of the interfacial roughness seen in Figure 2 might be due to the sample preparation and how much is intrinsic to the grown structures.
- Pits and porosity in the films are introduced with little or no introduction on page 11. The possible origin of these should be more carefully introduced ahead of the subsequent analysis.
- The authors claim a six-fold anisotropy in the angle-dependent FMR measurements of Fig. 12. While uniaxial anisotropy is clear to see in these plots, the evidence for 6-fold periodicity is not terribly convincing. This needs to be more strongly justified.
- 14 appears to show that that the 4nm Ge film exhibits 3D variable range hopping behavior. However, the temperature exponent of 65K is extremely small. If one assumes that the localisation length in Ge is ~ 1nm this corresponds to an extremely high density of localized states, much higher than that extracted in other studies of polycrystalline Ge films. Also, one would probably expect to see 2D VRH behavior (with an exponent of 1/3) rather than 3D behavior in such thin films. These temperature-dependent transport results need to be discussed more carefully.
Author Response
Reviewer 2
Comments and Suggestions for Authors
The authors present the results of a very careful growth and characterization study of Fe3+xSi1-x/Ge/Fe3+xSi1-x trilayer structures. These have potential as vertical magnetotransport elements and are of particular interest because of the high spin polarization of the magnetic electrodes. From a practical point of view the growth of these structures is challenging due to lattice mismatch and the different required growth temperatures of the Fe3+xSi1-x and Ge layers.
- Two samples have been very carefully structurally, electronically and magnetically characterized and some important conclusions drawn. I believe that the paper should be suitable for publication in Nanomaterials once the following points have been addressed in a revised manuscript.
Thank for your time to read the manuscript. We appreciate very much your positive feedback on our work.
- The very comprehensive characterization of two samples with different Ge thicknesses has been described in the manuscript. We are given to understand that these effectively represent two different limits, one corresponding to epitaxial growth and one corresponding to polycrystalline growth. While it would be unreasonable to expect the same level of detailed characterization for a much larger number of samples, do the authors have data (even quite qualitative data) on samples with other Ge thicknesses which demonstrate that these two are truly illustrative of the two limits?
We understand that all information we obtain while keeping experimenting with the growth of epitaxial trilayers based on Fe-Si ferromagnetic alloys with the intermediate Ge layer is possible to include in a single paper of a reasonable length. In our samples, series dedicated to this topic one can find variation of the Fe/Si ratio for upper and lower layers, variation of Fe-Si and Ge layer thicknesses, variation of growth temperature and additional post-annealing step, the use of Fe-enriched buffer layer between silicon substrate and Fe3Si layer to suppress silicon diffusion. When needed we prepare separate layers like it is described in the articles. In this case, we carried out additional analysis of the dislocation border by TEM. In general, when we are at the Fe-rich corner of Fe3+xSi1-x alloys the Ge layer is under tensile strain and it increases even more due to incorporation of Ge atoms into Fe3+xSi1-x. It is easily seen in Figure 4. Thus, without relaxation via termination of silicon-enriched layer of surface Fe3+xSi1-x the Ge layer becomes strained and tend to lower the total energy through the formation of 3D islands and their faceting. When the thicknesses of the Fe3+xSi1-x lower layer increases or it becomes more enriched additional azimuthal orientation appear. Further strain increase results in the additional texture on other plane that is Ge(111). As a result the upper layer becomes textured or polycrystalline. In this article we dwell on two limiting cases, which show the dramatic difference. Surely, a short review article on this is planned with fewer characterization methods, mainly the RHEED as a probe for the structural analysis. Here we highlight the importance of incorporation of Ge due to different thermal history of the sample. Indeed thick layer of Ge with high structural quality is only possible in narrow technological regime when stoichiometric composition Fe3Si is preserved with Si termination of Fe3Si(111) plane, which is was shown by Japanese colleagues we mentioned in our paper.
- The details of the structural analysis of the films are quite extensive and do not make very easy reading. The authors might want to consider moving some of the detail to an appendix and just retaining to key results in the main manuscript.
Group of authors of this work considered the MPDI Nanomaterials as excellent opportunity to publish lengthy manuscript with a detailed description of methods incorporated to the main body since the journal publish only the digital copies. Nevertheless, we understand that too much detail may make reading difficult and we divided the structural properties section into four smaller parts as follows.
3.1.1 Analysis of epitaxial orientation relationships
3.1.2 Estimation of lattice distortions
3.1.3 Characterisation of the element depth distribution
3.1.4 Surface morphology and dislocation characteristics
They are intended to guide reader through the section and easily skip unwanted reading.
- The authors should state what protective layer was used when fabricating the TEM samples, was it Pt? It would be useful if they could reflect on how much of the interfacial roughness seen in Figure 2 might be due to the sample preparation and how much is intrinsic to the grown structures.
To prepare samples for the TEM study we used the Hitachi FB-2100, a single-beam focused ion beam (FIB) system (without electron gun). Therefore, in order to avoid damage to the near-surface layers during mask deposition, samples are covered with a protective layer of amorphous SiO with a thickness of 200 nm. This layer is visible in the pictures. After FIB preparation samples were Ar+ ion-polished at low energies (~1 keV). Sample preparation introduces minimal distortions in the resulting images, therefore, we can say with great confidence that the observed interfacial roughness is intrinsic to the grown structures.
- Pits and porosity in the films are introduced with little or no introduction on page 11. The possible origin of these should be more carefully introduced ahead of the subsequent analysis.
We aimed at careful qualitative and quantitative analysis of porosity based on dislocation lattice appeared due to lattice misfits of epitaxial interface and Terrace-Ledge-Kink (TLK) mechanism, which is described in the article. We agree it is better to introduce the reader to this part of analysis. The following text was added to the main text.
A significant difference in the thin surface morphology indicates different formation mechanisms for the samples discussed. Variation of the thermal history of the samples results in different level of incorporation of Ge atoms into Fe3+xSi1-x layers and causes variation of residual stress. The observable surface morphological characteristics are to be analysed further.
- The authors claim a six-fold anisotropy in the angle-dependent FMR measurements of Fig. 12. While uniaxial anisotropy is clear to see in these plots, the evidence for 6-fold periodicity is not terribly convincing. This needs to be more strongly justified.
You are right, uniaxial anisotropy is visible in Fig. 12 with the naked eye for all five lines that were extracted from the two FMR spectra. Nevertheless, it is clearly seen from Fig. 12 (b), Fig. 12 (e) and Fig. 12 (f) there is some extra contribution to the uniaxial anisotropy. Considering this, during analysis, for solving the Landau – Lifshitz – Gilbert equation four-fold and six-fold symmetry contributions were added, which are characteristic of crystals of ferromagnetic iron silicides. Taking into account six-fold symmetry gives a better fit to the experimental curve than taking into account only four-fold symmetry. As a result for the line presented in Fig. 12 (b) we found that Hk6 is closer than Hk4. Moreover, Hk6 is only 2.7 times lower than uniaxial Hk2. Therefore, it was fair to consider additional contributions in our analysis. Based on these data, we concluded that, most likely, the observed features in the angular dependences of the FMR resonance field are associated with the crystal symmetry and the film/substrate orientation relationship.
- 14 appears to show that that the 4nm Ge film exhibits 3D variable range hopping behavior. However, the temperature exponent of 65K is extremely small. If one assumes that the localisation length in Ge is ~ 1nm this corresponds to an extremely high density of localized states, much higher than that extracted in other studies of polycrystalline Ge films. Also, one would probably expect to see 2D VRH behavior (with an exponent of 1/3) rather than 3D behavior in such thin films. These temperature-dependent transport results need to be discussed more carefully.
Indeed, we also expected for our samples semiconductor-like R (T) curves like (ln (R) ~ 1 / T) or 2D VRH behavior (with an exponent of 1/3) because of very thin films of Ge. However, analysis of R (T) showed that 2D VRH fitting is worse than 3D VRH. To check it directly we measured R (H) at 4 K in parallel and perpendicular magnetic field. If 2D transport dominate, we should see the difference in such experiment. Nevertheless, we found no difference between parallel and perpendicular magnetic field (Fig. A1), which is consistent with the conclusions drawn from the R (T) analysis. As a result, we cannot conclude on any significant 2D contribution to the mechanisms of charge carrier transport in our samples. We do not include R (H) in the article, since this would increase the already large article but it would not elucidate the mater deeper.
Small temperature exponent and corresponding localization length are most likely due to the small thickness of the Ge interlayer accordingly, the high density of defects. An extremely high density of localized states can be associated with island-like nature of Ge film. At small thickness process of growth of the films goes on by 2D islands. Then, upon reaching certain thickness it transforms into the 3D regime. According to the TEM we have continuous Ge films but from AFM data we found “pits”. Assuming this we can guess that we stopped the growth of Ge layer at very first stage of 3D growth regime that affects the transport properties drastically.
Additional discussion was added to the manuscript.
English language and style were slightly corrected and highlighted in yellow.
In the manuscript, all other changes are highlighted in green.

Reviewer 3 Report
The authors fabricated iron-rich Fe3+xSi1-x/Ge/Fe3+xSi1-x (0.2 < x < 0.64) heterostructures with various Ge thicknesses on a Si(111) surface by molecular beam epitaxy. The structural and morphological properties of the synthesized samples have been systematic studies. Their results are useful for researchers in this field. My particular concerns are:
- In Figure 11,very different HL loops have been observed for samples with 4 and 7 nm Ge. The origin for this difference should be discusses in details. Is there any RKKY interaction between two Fe3Si? How did you precisely control the thickness of Ge?
- The MR curves for the Fe3+xSi1-x/Ge/Fe3+xSi1-x (0.2 < x < 0.64) heterostructures should be supplied. Did you observed any spin valve effect?
- What is origin of the third FMR line appeared in the sample with the thicker Ge?
Author Response
Reviewer 3
Comments and Suggestions for Authors
- The authors fabricated iron-rich Fe3+xSi1-x/Ge/Fe3+xSi1-x (0.2 < x < 0.64) heterostructures with various Ge thicknesses on a Si(111) surface by molecular beam epitaxy. The structural and morphological properties of the synthesized samples have been systematic studies. Their results are useful for researchers in this field.
Thank you very much for your positive feedback on our work.
- In Figure 11,very different HL loops have been observed for samples with 4 and 7 nm Ge. The origin for this difference should be discusses in details.
Difference of hysteresis loops for samples with 4 and 7 nm Ge is due to difference between iron silicide layers in this two samples. In addition to evident distinction of upper Fe3+xSi1-x layers, lower Fe3+xSi1-x layers also can be different due to a two-times more extended exposition at elevated temperature for 7 nm Ge sample. As a result, from the structural analysis we assumed a more complex interface structure and composition at the lower Ge/Fe3+xSi1-x and Fe3+xSi1-x/Si boundaries that leads to decreasing of saturation magnetisation MS and other loops’ features. To extract quantitative difference between samples we preferred to analyse and discuss the FMR data.
Additional discussion was added to the manuscript.
- Is there any RKKY interaction between two Fe3Si?
No, we didn’t found magnetic coupling between Fe3Si in both samples neither by VSM nor by FMR. In particular, in the FMR spectra the crossing of lines was observed during sweeping of out-of-plane angle. It occurs because the films have different MS and corresponding shape anisotropy filed 4π·MS.
Additional discussion was added to the manuscript.
- How did you precisely control the thickness of Ge?
The deposition rate is calibrated in situ separately before the sample synthesis with help of the single-wave ellipsometry. This allows a calibration of the thicknesses at the center of a substrate in comparison with the quartz monitor.
- The MR curves for the Fe3+xSi1-x/Ge/Fe3+xSi1-x (0.2 < x < 0.64) heterostructures should be supplied. Did you observed any spin valve effect?
At this moment we systematically measured R(H) dependences for only upper films (electrical contacts on upper silicide films) and for Ge films (electrical contacts on upper film with etched gap as indicated on Fig.2). For FM upper films we observed ordinary anisotropic MR effect as expected. Fabricated simple device for studying transport properties of Ge in our 3-layered structure has 500 µm gap in FM film. For such a device there are no expectation to observe spin valve effect. We found only small negative MR at temperature of 4 K that can be associated with weak localization effect. Now we are developing technological process to create micron-scale devices, which will make possible investigation of spin-dependent transport in our vertical structures.
- What is origin of the third FMR line appeared in the sample with the thicker Ge?
At this research stage it is hard to determine the specific origin of the 3rd FMR line in the sample #7. Most likely it can be associated with the presence of additional ferromagnetic phases ordisordered A2 Fe-Ge-Si alloys formed near any of 3 interfaces Si/ Fe3+xSi1-x or Fe3+xSi1-x/Ge or Ge/ Fe3+xSi1-x.
English language and style were slightly corrected and highlighted in yellow.
In the manuscript, all other changes are highlighted in green.

Round 2
Reviewer 2 Report
The authors have satisfactorily addressed the issues raised in my report in a revised manuscript. I can now recommend publication in its current form.
Reviewer 3 Report
The authors addressed most of my concerns. I suggest to publish as is.